# Hyperbilirubinemia in Gunn Rats Is Associated with Decreased Inflammatory Response in LPS-Mediated Systemic Inflammation

**DOI:** 10.3390/ijms20092306

**Published:** 2019-05-09

**Authors:** Petra Valaskova, Ales Dvorak, Martin Lenicek, Katerina Zizalova, Nikolina Kutinova-Canova, Jaroslav Zelenka, Monika Cahova, Libor Vitek, Lucie Muchova

**Affiliations:** 1Institute of Medical Biochemistry and Laboratory Diagnostics, First Faculty of Medicine, Charles University and General University Hospital in Prague, 12108 Prague, Czech Republic; petra.valaskova@lf1.cuni.cz (P.V.); aleshdvorak@gmail.com (A.D.); mleni@centrum.cz (M.L.); katka.ziza@seznam.cz (K.Z.); vitek@cesnet.cz (L.V.); 2Institute of Pharmacology, First Faculty of Medicine, Charles University and General University Hospital in Prague, 12800 Prague, Czech Republic; Nikolina.Canova@lf1.cuni.cz; 3Department of Biochemistry and Microbiology, University of Chemistry and Technology, 16628 Prague, Czech Republic; jar.zelenka@gmail.com; 4Department of Experimental Diabetology, Institute of Clinical and Experimental Medicine, 14021 Prague, Czech Republic; moca@ikem.cz; 54th Department of Medicine—Department of Gastroenterology and Hepatology, First Faculty of Medicine, Charles University and General University Hospital in Prague, 12808 Prague, Czech Republic

**Keywords:** bilirubin, Gunn rats, hyperbilirubinemia, inflammation, LPS, NF-κB

## Abstract

Decreased inflammatory status has been reported in subjects with mild unconjugated hyperbilirubinemia. However, mechanisms of the anti-inflammatory actions of bilirubin (BR) are not fully understood. The aim of this study is to assess the role of BR in systemic inflammation using hyperbilirubinemic Gunn rats as well as their normobilirubinemic littermates and further in primary hepatocytes. The rats were treated with lipopolysaccharide (LPS, 6 mg/kg intraperitoneally) for 12 h, their blood and liver were collected for analyses of inflammatory and hepatic injury markers. Primary hepatocytes were treated with BR and TNF-α. LPS-treated Gunn rats had a significantly decreased inflammatory response, as evidenced by the anti-inflammatory profile of white blood cell subsets, and lower hepatic and systemic expressions of IL-6, TNF-α, IL-1β, and IL-10. Hepatic mRNA expression of LPS-binding protein was upregulated in Gunn rats before and after LPS treatment. In addition, liver injury markers were lower in Gunn rats as compared to in LPS-treated controls. The exposure of primary hepatocytes to TNF-α with BR led to a milder decrease in phosphorylation of the NF-κB p65 subunit compared to in cells without BR. In conclusion, hyperbilirubinemia in Gunn rats is associated with an attenuated systemic inflammatory response and decreased liver damage upon exposure to LPS.

## 1. Introduction

Bilirubin (BR), the end product of the heme degradation pathway in the intravascular compartment, is an important endogenous antioxidant, and it plays a crucial role in protection against oxidative stress as has been demonstrated in numerous in vitro, in vivo, and clinical studies (for review, see [1]). Recently, it has been shown that BR exerts potent anti-inflammatory and immunomodulatory activities [2]. In fact, mild hyperbilirubinemia has been associated with a reduced risk of diseases linked to increased oxidative stress and chronic inflammation (for review, see [3]).

A wide array of BR anti-inflammatory effects are mediated by multiple mechanisms, and indeed, BR is capable of modulating all stages of both the innate as well as the adaptive immune system [2]. These, predominantly suppressing activities, are aimed against: the complement system [4], damage-associated molecular patterns (DAMPs) signaling [5], Toll-like receptors (TLRs), such as TLR4 (a bacterial lipopolysaccharide (LPS) receptor) [6], macrophage activities [7] as well as B cell-mediated antibody production [8], and differentiation of T cells, including regulatory T cells (Tregs) [9], all with wide-spread potential clinical consequences towards autoimmune diseases [10] and transplant medicine [5,9].

An increasing body of evidence suggests that mildly elevated BR concentrations could suppress production of pro-inflammatory cytokines [5,10,11]. The secretion of cytokines is under the control of nuclear factor kappa B (NF-κB), a master regulator of numerous genes involved in the immune and inflammatory responses [12]. In the canonical pathway, NF-κB is activated by many signals including bacterial LPS, which binds to the LPS-binding protein (LBP), and then interacts with TLR4/CD14 receptors [13]. In resting cells, NF-κB is inactive, located in the cytoplasm bound to its inhibitor IκB. Upon activation, the IκB kinase (IKK) complex activates NF-κB by phosphorylating IκB, resulting in ubiquitination and proteasome degradation of IκB. Active NF-κB then translocates into the nucleus and activates specific genes [14]. Taking into consideration the reported inhibitory effects of BR on protein phosphorylation [15] as well as its general immune system-suppressing activities [10], we hypothesize that BR might also interfere with phosphorylation of NF-κB p65 subunit, and thus prevent translocation of NF-κB into the nucleus.

Therefore, the aim of our study was thus to evaluate the pathophysiological role of BR in LPS-induced inflammation in hyperbilirubinemic Gunn rats and primary hepatocytes isolated from hyper- and normobilirubinemic animals.

## 2. Results

### 2.1. Hyperbilirubinemia in Gunn Rats Is Associated with Decreased Systemic Inflammatory Response in LPS-Induced Sepsis

To evaluate the effect of BR on systemic and hepatic inflammation, the complete blood count, as well as serum markers of liver injury, was measured in hyperbilirubinemic Gunn rats as well as in their normobilirubinemic heterozygous littermates. Interestingly, higher white blood cell (WBC) counts were observed after LPS treatment in control rats as compared to in hyperbilirubinemic Gunn animals ((12.39 ± 5.26) × 10^9^/L vs. (8.70 ± 1.94) × 10^9^/L, *p* = 0.05). Following LPS administration, significant increases were detected in the proportions of neutrophils (396 ± 301%, *p* < 0.01), monocytes (565 ± 242%, *p* < 0.01), basophils (338 ± 271%, *p* < 0.05), as well as eosinophils (448 ± 419%, *p* < 0.05), together with a decrease in the lymphocyte count (up to 23 ± 13%, *p* < 0.01) in control animals. However, these changes were substantially attenuated in hyperbilirubinemic Gunn rats (Figure 1a–f).

Simultaneously, marked changes in the CD4^+^/CD8^+^ T cells were observed in both hyperbilirubinemic Gunn rats and control animals upon exposure to LPS. In fact, the CD4^+^/CD8^+^ T ratio, a marker of immune activation [16], was 13 times higher in hyperbilirubinemic Gunn rats as compared to in controls (*p* < 0.05) (Figure 1g,h).

To evaluate the effect of hyperbilirubinemia on mediators of systemic inflammation, we first measured mRNA expression of the selected cytokines in the liver tissue as well as in the WBC of control and LPS-treated animals. The lower expressions of liver pro-inflammatory cytokines interleukin-6 (*IL-6*) (50 ± 49%, *p* < 0.05) and tumor necrosis factor-α (*TNF-α*) (59 ± 26%, *p* < 0.05) were observed in Gunn rat livers without LPS treatment compared to those in heterozygous littermates. After LPS administration, significantly lower increases in pro-inflammatory *TNF-α* (34 ± 21%, *p* < 0.05), interleukin-1β (*IL-1β*) (57 ± 30%, *p* < 0.05), and anti-inflammatory interleukin-10 (*IL-10*) (40 ± 22%, *p* < 0.05, Figure 2a–d) were detected in Gunn rats as compared to in normobilirubinemic controls 12 h after saline or LPS administration. Similar results in mRNA cytokine expressions were observed also in the WBC. Indeed, the elevation levels of cytokines *IL-6*, *TNF-α*, *IL-1β* and *IL-10* after LPS administration were significantly attenuated in Gunn rats (49 ± 35%, 43 ± 43%, 31 ± 28%, and 24 ± 13%, respectively, *p* < 0.05) compared to that in control animals (Figure 2e–h).

Serum concentrations of selected cytokines were measured to confirm the functional translation of their mRNA expressions. In untreated animals, the concentrations of all tested cytokines were under the limit of detection. However, after LPS treatment, the changes in concentrations of most cytokines followed the pattern of mRNA expressions (although the concentration of IL-1β was under the limit of detection). Compared to that of controls, lower concentrations of IL-6 (35 ± 1%) as well as those of TNF-α (60 ± 56%) and IL-10 (25 ± 23%, *p* < 0.05) were observed in Gunn rats exposed to LPS (Figure 3). This data resulted in a marked difference in the IL-10/TNF-α ratio, a marker of immune homeostasis, between H LPS+ and G LPS+ experimental groups (0.51:0.19, *p* < 0.05).

Since the response of an organism to LPS sepsis involves production of LBP, an acute phase protein, by the liver, we tested in whether hyperbilirubinemia might affect production of this mediator. Indeed, *LBP* mRNA expression was upregulated in the liver of Gunn rats compared to in their normobilirubinemic littermates both before (142 ± 37%, *p* < 0.05) and after LPS treatment (148 ± 48%, *p* < 0.05, Figure 4a). Based on the results from in vivo experiments, *LBP* expression in primary hepatocytes was assessed upon exposure to LPS. The expression of *LBP* gradually increased starting at 6 h in Gunn primary hepatocytes exposed to 20 and 40 µM BR (*p* < 0.05, Figure 4b). Interestingly, no significant increase in mRNA expression of *LBP* upon incubation with BR was observed in control hepatocytes (Appendix A).

Importantly, markers of liver injury such as alanine transaminase (ALT) and aspartate transaminase (AST) activities were lower in the LPS-treated Gunn rats compared to in LPS-treated controls (1.87 ± 1.14 vs. 5.55 ± 3.32 µkat/L, and 4.28 ± 2.26 vs. 6.22 ± 2.88 µkat/L, respectively, *p* < 0.05 for both comparisons, Figure 5a,b).

### 2.2. Pretreatment of Primary Hepatocytes with Bilirubin Protects against Inflammation-Induced Cell Death

To assess the underlying mechanisms of anti-inflammatory effect of BR, primary hepatocytes isolated from hyperbilirubinemic Gunn rats and normobilirubinemic heterozygous controls were used for in vitro experiments. Both types of primary liver cells were exposed to BR or/and TNF-α to find out whether constitutive/basal BR could have protective effects on cell viability. No differences in intracellular BR levels were observed between primary hepatocytes isolated from hyperbilirubinemic and normobilirubinemic animals independently of BR treatment (Appendix A). Nevertheless, primary hepatocytes isolated from hyperbilirubinemic Gunn rats were more resistant to TNF-α-induced cell death as compared to in the control cells (16 ± 10%, *p* < 0.05, Appendix A), consistent with in vivo data on the effect of hyperbilirubinemia on the liver injury markers described above.

### 2.3. Effect of Bilirubin on NF-κB Pathway

To examine the role of BR in regulation of NF-κB, a key mediator of inflammatory signaling, we investigated whether BR pre-treatment affects TNF-α-mediated NF-κB activation. Both types of primary hepatocytes were pre-treated with 10–40 µM BR and then exposed to TNF-α. As expected, TNF-α resulted in an increased phosphorylation of the NF-κB p65 subunit. Importantly, pretreatment with BR significantly decreased TNF-α-induced NF-κB p65 subunit phosphorylation (Figure 6a) (*p* < 0.05). No significant changes were detected in total levels of NF-κB p65 protein (Figure 6b), IKKβ protein, and inhibitor IκBα, as well as in phosphorylation of IKKα/β and IκBα after BR and TNF-α treatment (Appendix A). Interestingly, only BR itself increased phosphorylation of IKKα/β (Appendix A).

## 3. Discussion

BR has been shown to be an important cytoprotective and especially antioxidant molecule at physiological or mildly elevated concentrations [1,3]. Even though in vitro studies as well as clinical observations suggest that BR might also possess considerable anti-inflammatory properties [2], surprisingly scarce data have been published on hyperbilirubinemic animal models of inflammation. In our study, we used a model of LPS-induced sepsis in Gunn rats with plasma-unconjugated bilirubin levels at around 60 µmol/L (compared to heterozygotes with 2 µmol/L).

Interestingly, a marked attenuation of WBC pro-inflammatory response with decreased counts of neutrophils and monocytes was observed after LPS treatment in hyperbilirubinemic Gunn animals, accompanied with substantial changes in the CD4^+^/CD8^+^ T cell ratio, an important marker of immune activation [16]. The expansion of CD8^+^ cells is also driven by the activity of NADPH oxidase (NOX2) [17]. Therefore, the beneficiary CD4^+^/CD8^+^ ratio observed in our study could at least partially be due to a previously reported inhibitory effect of BR on NOX2 activity [18]. 

The major driving force of neutrophil mobilization from bone marrow and other hematopoetic compartments during sepsis are pro-inflammatory cytokines, of which generation was significantly attenuated in our hyperbilirubinemic rats. Since an overabundance of neutrophils during severe inflammation might have serious damaging effects [19], its amelioration seems to contribute to hyperbilirubinemia-induced protection. In addition, overwhelmed cytokine production during sepsis is also considered detrimental. In fact, lower mortality was observed in rats exposed to LPS and treated with a monoclonal antibody against TNF-α [20], and beneficiary effects were also observed for anti-IL-1β as well as anti-IL-6 antibody treatment in other experimental models of sepsis [21,22].

Together with the decreased expression of pro-inflammatory cytokines observed in our septic hyperbilirubinemic animals, there was also reduced production of IL-10. Although IL-10 is generally considered to be an anti-inflammatory cytokine, its overproduction might also be harmful and result in immunosuppression [23]. In fact, serum IL-10 concentrations were demonstrated in a human clinical study to correlate well with the sepsis severity and mortality, as also did the high IL-10/TNF-α ratio [24]. Thus, the balance between IL-10 and TNF-α seems to be important for immune homeostasis maintenance, as demonstrated by a curative effect of blocking of the IL-10 pathway in several models of bacterial infections such as from *Listeria* [25], *Klebsiella* [26], *Pseudomonas* [23], *Streptococcus* [27], or *Mycobacterium* [28]. It has been suggested that this approach seems to be promising as an adjunct therapy for severe septicemias. In our study, the IL-10/TNF-α ratio was markedly lower in hyperbilirubinemic Gunn rats, consistent with the better survival rate in Gunn rats exposed to LPS observed in the previous study by Lanone et al. [29]. In concordance with these observations, significantly lower values of hepatocellular liver injury markers were observed in Gunn rats exposed to LPS, as similarly in previous studies [18,29]. In addition, we have previously shown that BR protects the liver against pro-oxidative effects of elevated bile acids in cholestasis [30], further emphasizing the role of BR in hepatoprotection.

To find the factors contributing to a decreased BR-mediated inflammatory response in Gunn rats to LPS, we investigated the expression of LBP in the liver tissue of our experimental animals. LBP, a plasma protein mainly produced by hepatocytes, plays a crucial role in LPS recognition and signaling and is considered an important mediator of the inflammatory reaction [31]. It binds LPS in plasma and transports it via cluster of differentiation 14 (CD14) to the Toll-like receptor 4 (TLR4)/MD-2 signaling complex triggering a range of pro- and anti-inflammatory responses. Dysregulation of this finely tuned signaling cascade could result in a deleterious effect on organism including sepsis and septic shock [32]. Even though the role of LBP in the activation/inhibition of the inflammatory response is probably a dual one, depending on its serum concentration, it has been described that high LBP levels inhibit LPS-mediated cytokine release and prevent hepatic failure in vivo [33]. In our study, hepatic *LBP* expression was significantly higher in Gunn rats before and after LPS treatment, suggesting a role of hyperbilirubinemia in LBP-mediated LPS signaling. Moreover, the treatment of Gunn primary hepatocytes with BR resulted in an increased *LBP* expression, indicating that BR might affect LBP production, and thus contribute to an attenuated inflammatory response in hyperbilirubinemic subjects.

The production of pro-inflammatory cytokines during sepsis leads to activation of NF-κB [34]. In fact, our experiments on primary rat hepatocytes demonstrated that BR exposure resulted in decreased phosphorylation of the p65 subunit of the NF-κB protein complex, a phenomenon which might be related to both general inhibitory effects of BR on protein phosphorylation [15], as well as inhibition of phosphorylation via suppressed TNF-α signaling [35]. It is thus likely that anti-inflammatory and cytoprotective effects of BR may at least in part be due to attenuation of NF-κB-driven transcription. On the contrary, we did not observe any inhibition of IκB phosphorylation, which has previously been reported, but at much higher BR levels [10]. Moreover, phosphorylation of IKK was increased by BR itself. Our data are consistent with previous reports demonstrating this specific inhibitory effect of both BR [36] and biliverdin [37,38]. It is also interesting to note that the CD4^+^/CD8^+^ ratio (similarly to Tregs and myelopoesis), which changes in our hyperbilirubinemic rats, is also regulated by the activity of NF-κB [39]. 

Furthermore, Gunn rat hepatocytes were more resistant to TNF-α-induced cytotoxicity, although no changes in intracellular BR concentrations were observed compared to control cells. These data suggest that not only BR itself, but also “bilirubin priming”, triggering adaptive, para-hormetic mechanisms under hyperbilirubinemic conditions, might significantly contribute to the observed hepatoprotection; however, intensive research is needed to identify these mechanisms.

## 4. Materials and Methods

### 4.1. Chemicals and Reagents

BR, the bovine serum albumin (BSA), rat TNF-α, LPS from Escherichia coli 0114:B4, human insulin solution, Williams’ E Medium, Collagen type I from rat tail tendon, 2,6-di-tert-butyl-4-methylphenol (BHT), Thiazolyl Blue Tetrazolium Bromide (MTT), RNAlater, and tetrabutyl-ammonium hydroxide (TBA, 40 % in water) were purchased from Sigma-Aldrich (St. Louis, MO, USA); the chloroform (HPLC grade), methanol (HPLC grade), *n*-hexane, ethyl acetate, and acetonitrile were purchased from Merck (Darmstadt, Germany); and 4×-Laemmli sample buffer was from Bio-Rad (Hercules, CA, USA). 

As described earlier, the BR was purified before use [40]. For the experiments, BR (2.8 mg) was dissolved in 2 mL of 0.1 M NaOH and immediately mixed with 1 mL of 0.1 M phosphoric acid. The mixture was diluted with a BSA solution (660 µM BSA in 25 mM phosphate buffer, pH: 7.7) to reach a final concentration of 480 µM BR in a phosphate buffer and then serially diluted with a BSA solution to yield solutions with final BR concentrations within the range of 10–40 µM.

### 4.2. In Vivo Studies

Hyperbilirubinemic adult female Gunn rats and their normobilirubinemic heterozygous littermates (*n* range: 8–25 per group, weight range: 160–260 g) had access to water and food ad libitum. Gunn rats were kindly provided by Cluster in Biomolecular Medicine (University of Trieste, Italy). All protocols were approved by the Animal Research Committee of the 1st Faculty of Medicine, Charles University, project No. MSMT-25538/2018-2 (29 August 2018) as well as by the Institute of Molecular Genetics of the Academy of the Sciences of the Czech Republic, project No. PP 67/2018 (24 July 2018) and carried out in accordance with the Guide for the Care and Use of Animals of the National Institutes of Health.

The rats were divided into 4 groups: normobilirubinemic heterozygote (a) and hyperbilirubinemic Gunn (b) experimental groups were treated with LPS (H LPS+/G LPS+, 6 mg/kg intraperitoneally); and normobilirubinemic heterozygote (c) and hyperbilirubinemic Gunn (d) control groups received vehicle (saline). After 12 h, the animals were anesthetized (xylazin: 16 mg/kg, i.m.) and sacrificed. Blood for further biochemical analyses was obtained from the inferior vena cava and from the aorta for flow cytometry. Relevant organs (liver, heart, lung, kidney, spleen, and brain) were harvested, washed with ice-cold PBS, snap frozen in liquid nitrogen and stored at −80 °C. For RNA analysis, 100 mg of fresh liver was immediately placed in 2-mL microfuge tubes containing RNAlater and stored according to the manufacturer’s instructions until analysis.

### 4.3. Determination of Complete Blood Count with Differential and Serum Biochemical Markers 

Complete blood counts were measured from whole blood using an XN-1000™ automatic analyzer (Sysmex, Lincolnshire, IL, USA). Serum biochemical markers (ALT and AST activities) were determined by standard assays using an automatic analyzer (Modular analyzer, Roche Diagnostics GmbH, Mannheim, Germany). 

### 4.4. Determination of Serum Cytokine Concentrations

Determination of serum cytokine concentrations were performed using commercial rat ELISA kits (Duo-Sets kits for IL-1β/IL-1F2, IL-10, TNF-α, and IL-6; Bio-Techne R&D Systems, Minneapolis, MN, USA) according to the manufacturer’s instructions. 

### 4.5. Flow Cytometry of Lymphocytes 

Blood samples (250 µL of whole blood) were collected in tubes with 3% potassium EDTA. After lysis of the red blood cells (twice) using ACK buffer (0.15 M NH4Cl, 10 mM KHCO3, and 1 mM EDTA monosodium; pH: 7.3) for 15 and 5 min separately, followed by washing with PBS (twice), the cells were simultaneously stained for effector T cells. Cells were surface-stained using the following anti-rat antibodies: anti-CD45-FITC (OX-1, Thermo Fisher Scientific, Waltham, MA, USA), anti-CD4-BV-786 (OX-35, BD Biosciences, San Jose, CA, USA), anti-CD8α-PerCP-e710 (OX-8, Thermo Fisher Scientific), and anti-CD62L-PE (OX-85, SONY) for CD4/CD8 T cells panel. The cell suspensions were analyzed by flow cytometry (BD LSR II including an FACSFlow Supply of the High Throughput Sampler System, BD Biosciences). 

### 4.6. Gene Expression Analyses

Total RNA from liver tissue and blood was isolated using a GenUP™ Total RNA Kit (Biotechrabbit GmbH, Hennigsdorf, Germany) and a Total RNA Mini Kit (Geneaid Biotech Ltd, New Taipei City, Taiwan), respectively. The quantity and purity of isolated RNA were evaluated spectrophotometrically. cDNA was generated by a High-Capacity cDNA Reverse Transcription Kit (Thermo Fisher Scientific) and stored at −20 °C until analysis. Quantitative real-time PCR was performed using TaqMan^®^ Fast Advanced Master Mix and a TaqMan^®^ Gene Expression Assay Kit for the following genes: *IL-6* (Rn01410330_m1), *TNF-α* (Rn99999017_m1), *IL-10* (Rn00563409_m1), *IL-1β* (Rn00580432_m1), *LBP* (Rn00567985_m1), and a rat endogenous control β-2 microglobulin (Rn005608865_m1). Results were expressed as the % of controls.

### 4.7. Primary Rat Hepatocyte Culture

Primary hepatocytes were isolated from anaesthetized Gunn and heterozygote (*n* = 3 each, weight range: 200–220 g) rats by two-step collagenase perfusion according to a published protocol [41]. Cell viability ranged from 75% to 85% (as trypan blue staining). Hepatocytes were further diluted to 0.8 million cells/mL with William’s E medium, supplemented with 1% penicillin/streptomycin, 1% L-glutamine, 0.06% insulin, and 5% fetal bovine serum. Primary hepatocytes were dispensed into a collagen-coated cell culture Petri dishes, 6-well and 96-well plates and allowed to attach for 3 h at 37 °C with 5% CO_2_ in the incubator. Unattached cells were removed after 3 h and a new medium was added. The following day, hepatocytes were cultured with complete culture medium containing BR (10–40 µM) and TNF-α (12–100 ng/mL) for 24 h.

### 4.8. Determination of Cell Viability and Intracellular Bilirubin Levels 

All experiments were performed under dim light to minimize BR degradation. Cell viability of primary hepatocytes seeded in 96-well plates was measured using an MTT test after 24 h incubation. Primary hepatocytes harvested from 10 cm Petri dishes were used for determination of intracellular BR as described previously [42].

### 4.9. Western Blot Analysis

Primary hepatocytes were lysed using a lysis buffer (5 M NaCl, 1 M Tris, pH = 8, 10% Triton-X 100), sonicated for 5 s and centrifuged at a speed of 14,000× *g* for 10 min (temperature: 4 °C). Supernatants (35–40 µg of protein) were diluted with a loading buffer (4× Laemmli Sample buffer, Bio-Rad, USA), denatured at 95 °C for 10 min, and separated by SDS-PAGE electrophoresis (10%). Proteins were transferred to a nitrocellulose membrane, blocked in 5% BSA in TTBS for 1.5 h and then incubated overnight at 4 °C with primary antibodies anti phospho-NF-κB p65 (Ser536) (dilution, 1:2000 *v*/*v*), anti NF-κB p65 (dilution, 1:3500 *v*/*v*), anti IκB-α (dilution, 1:3500 *v*/*v*), anti phospo-IκB-α (Ser132) (dilution, 1:1500 *v*/*v*), anti IKKβ (dilution, 1:3500 *v*/*v*), anti phospho-IKKα/β (Ser176/180) (dilution, 1:1500 *v*/*v*), as well as anti β-actin (dilution, 1:5000 *v*/*v*) as a loading control (all antibodies were from Cell Signaling Technology, Danvers, MA, USA). After being washed in TTBS buffer, membranes were incubated with swine anti-rabbit IgG-HRP secondary antibody (Dako, Glostrup, Denmark) and visualized using an ECL kit (LumiGLO^®^, Cell Signaling Technology). A Fusion Fx7 device and Bio-2D software (Vilber Lourmat, Collegien, France) were used to quantify the signals. Results were normalized to β-actin.

### 4.10. Statistical Analysis

Student parametric unpaired and paired t-tests were used for comparison of normally distributed data. Non-normally distributed data were analyzed with the Mann–Whitney rank sum test. Group mean differences were analyzed by ANOVA and Kruskal–Wallis tests. Depending on their normality, data are expressed as the mean with SD or the median with interquartile range. Differences were considered statistically significant when *p* < 0.05. Analyses were performed using GraphPad Prism 5.0 statistical software (GraphPad Software, Inc., San Diego, CA, USA).

## 5. Conclusions

In conclusion, hyperbilirubinemia in Gunn rats is associated with an attenuated systemic inflammatory response and decreased liver damage upon exposure to LPS, an effect associated with a modulation of innate immunity together with decreased production of pro-inflammatory cytokines and NF-κB activation. 

## Figures and Tables

**Figure 1 ijms-20-02306-f001:**
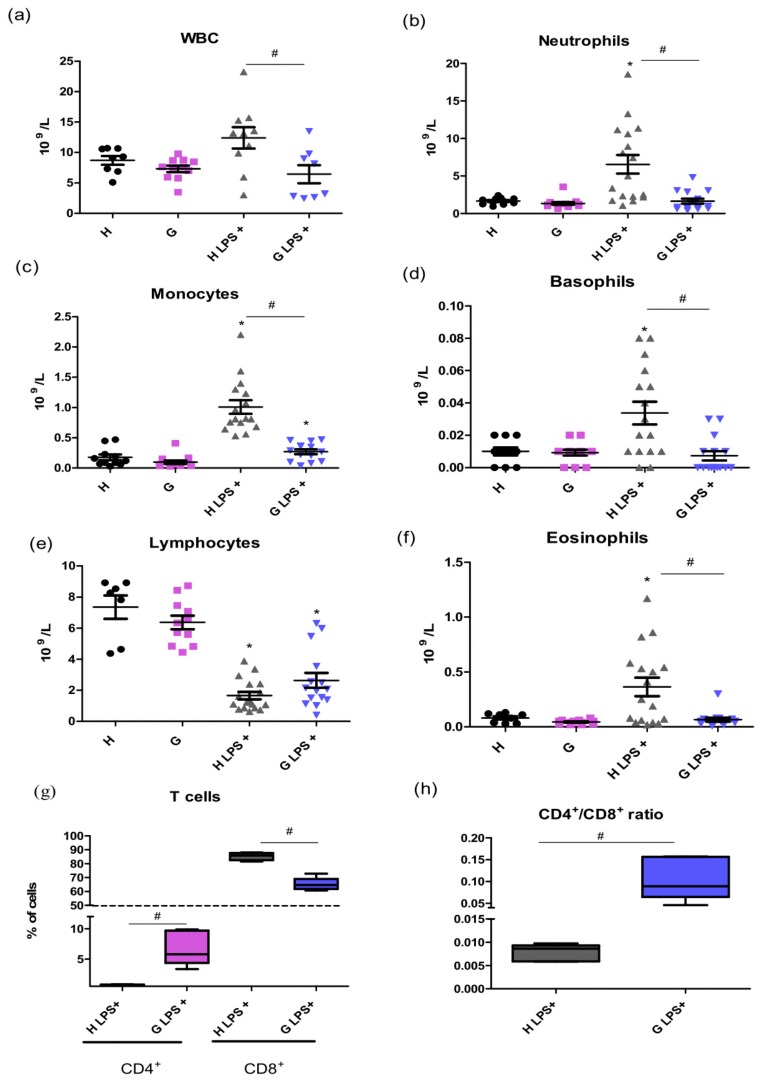
The effect of LPS-induced inflammation on WBC in hyperbilirubinemic Gunn rats. Total WBC cells (**a**) and their subpopulations (**b**–**f**) including T cells count (**g**) and CD4^+^/CD8^+^ ratio (**h**) were measured 12 h after LPS administration (6 mg/kg i.p.) in normobilirubinemic heterozygous controls (H or H LPS+) and hyperbilirubinemic Gunn rats (G or G LPS+), respectively. * *p* < 0.05 vs. corresponding control, # *p* < 0.05 vs. LPS-treated group. *n* = 8 animals per group (minimum).

**Figure 2 ijms-20-02306-f002:**
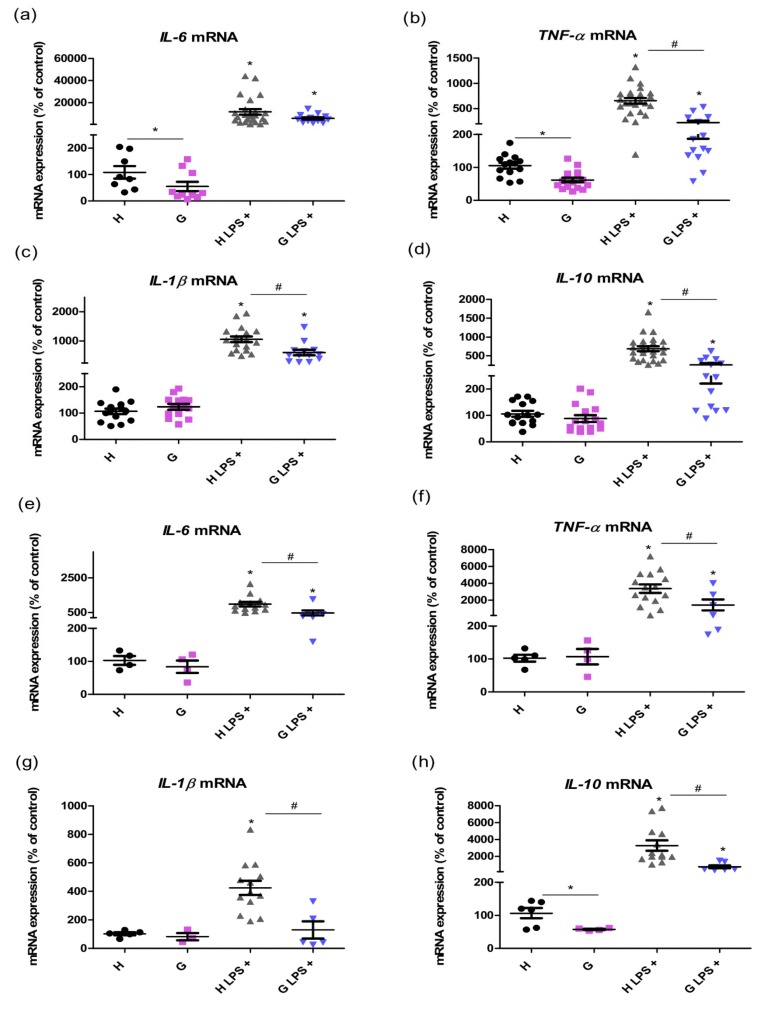
The effects of LPS-induced inflammation on mRNA cytokine expression in the liver and WBC of hyperbilirubinemic Gunn rats. mRNA expressions of pro- and anti-inflammatory cytokines *IL-6*, *TNF-α*, *IL-1β*, and *IL-10* were measured in the liver tissue (**a**–**d**) and white blood cells (**e**–**h**) 12 h after saline or LPS administration (6 mg/kg i.p.) in normobilirubinemic heterozygous controls (H or H LPS+) and hyperbilirubinemic Gunn rats (G or G LPS+), respectively. * *p* < 0.05 vs. corresponding control, # *p* < 0.05 vs. LPS-treated group. *n* = 5 animals per group (minimum).

**Figure 3 ijms-20-02306-f003:**
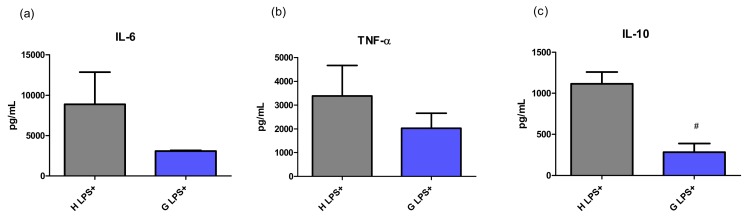
The effect of LPS-induced inflammation on cytokine concentration in serum of hyperbilirubinemic Gunn rats. Concentrations of pro-inflammatory cytokines IL-6, TNF-α, and anti-inflammatory IL-10 were measured 12 h after LPS administration (6 mg/kg i.p.) in normobilirubinemic heterozygous controls (H LPS+) and hyperbilirubinemic Gunn rats (G LPS+), respectively. # *p* < 0.05 vs. LPS-treated group. *n* = 5 animals per group (minimum).

**Figure 4 ijms-20-02306-f004:**
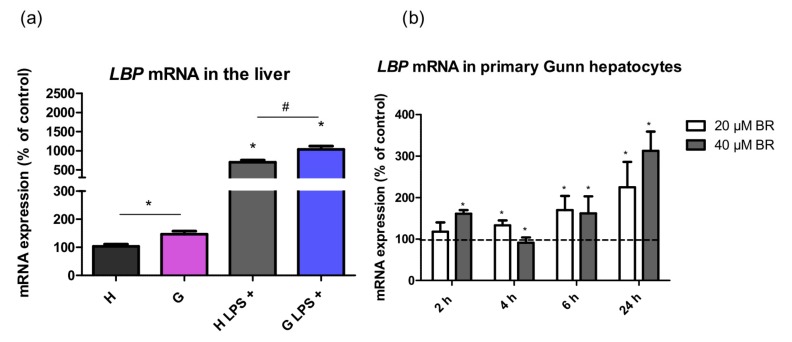
The effects of hyperbilirubinemia on lipopolysaccharide binding protein (*LBP*) mRNA expression in the liver tissues upon exposure to LPS (6 mg/kg i.p.) and in primary hepatocytes. mRNA expression of *LBP* was measured in the liver tissue (**a**) of normobilirubinemic heterozygous controls (H or H LPS+) and hyperbilirubinemic Gunn rats (G or G LPS+), respectively, and in primary hepatocytes (**b**). Primary hepatocytes isolated from Gunn rats were incubated with BR (20 and 40 µM) for 2, 4, 6, and 24 h. (**b**) Values are expressed as % of untreated control cells (100%). * *p* < 0.05 vs. controls, # *p* < 0.05 vs. LPS-treated group. (**a**) *n* = 12 animals per group (minimum); (**b**) *n* = 6 independent cell cultures per group.

**Figure 5 ijms-20-02306-f005:**
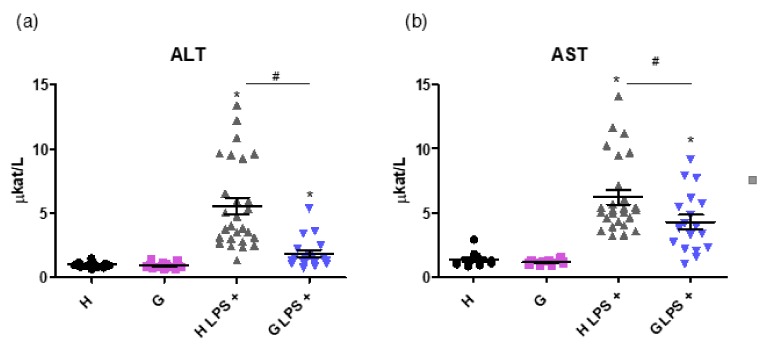
The effect of hyperbilirubinemia and inflammation on markers of the liver injury. ALT (**a**) and AST (**b**) activities, markers of liver injury, were measured in normobilirubinemic heterozygous controls (H or H LPS+) and hyperbilirubinemic Gunn rats (G or G LPS+) 12 h after saline or LPS administration (6 mg/kg i.p.), respectively. * *p* < 0.05 vs. corresponding control, # *p* < 0.05 vs. LPS-treated group. *n* = 8 animals per group (minimum).

**Figure 6 ijms-20-02306-f006:**
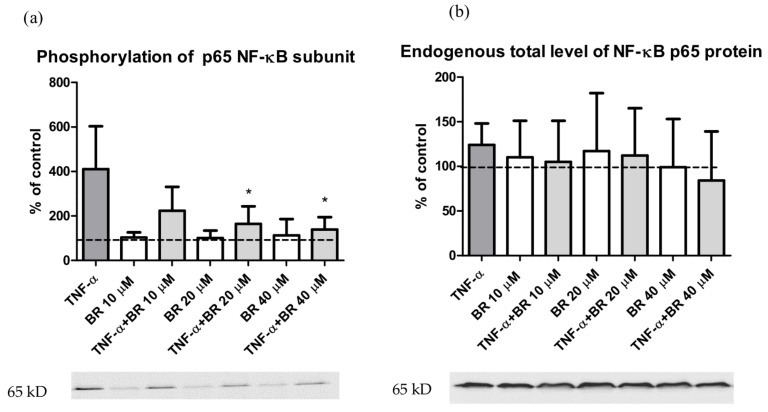
The effect of bilirubin on NF-κB p65 subunit phosphorylation. Both types of primary hepatocytes were pre-incubated with BR (0–40 µM) for 2 h and then treated with TNF-α (12 ng/mL) for 5 min. Phosphorylated (**a**) and total (**b**) NF-κB p65 subunits were measured by the western blot. Values are expressed as % of untreated control cells (100%). * *p* < 0.05 vs. TNF-α. *n* = 6 independent cell cultures per group.

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
