# Peer review of "Hyperbilirubinemia in Gunn Rats Is Associated with Decreased Inflammatory Response in LPS-Mediated Systemic Inflammation"

_ijms, 2019, doi:10.3390/ijms20092306_

Reviewer 1 Report

This study shows the protective effects of hyperbilirubinemia in Gunn rats on LPS-mediated inflammation and liver damage. I have some minor comments.

Full name of abbreviated proteins such as TNF-a, IL-1b and IKK etc. is required.

n number for each experiment including supplementary figures is necessary. Describe it in Figure legend. (e.g. n=3 independent cultures or n=12 animals)

The authors mentioned Fig. 4a (Line 127), Fig. 5a-b (line 130), and Fig. 4b (line 185). The order of Figure description is better if the authors demonstrate it orderly (i.e. Fig. 4a -> Fig. 4b -> Fig. 5a-b).

In Fig. 4b, Fig. 6a-b, and supplementary Figures, what dose 100% represent?

In Sup. Fig. S3b-d, the western blot data does not seem to match with Graph. In Fig. S3d, the first line of band is being cut (not complete band).

Figure legend for Fig. S3d: [Supp. Fig. S3. The effect of bilirubin on NF-κB signaling pathway. Primary hepatocytes were pre-incubated with BR (0-40 µM) for 2 h and then treated with TNF-α (12 ng/ml) for 5 min. Total (b) IκBα, (d) IKKβ and phosphorylated (a) IκBα and (b) IKKα/β were measured by Western blot.]. There is no (c) explanation.

Line 343: CO2 -> CO2

Author Response

Reviewer #1:

Thank you for your positive review and helpful comments. Below see our point-by-point response.

Responses to Reviewer #1:

This study shows the protective effects of hyperbilirubinemia in Gunn rats on LPS-mediated inflammation and liver damage. I have some minor comments.

Full name of abbreviated proteins such as TNF-a, IL-1b and IKK etc. is required. We added the full names into the text and also into the list of abbreviations

n number for each experiment including supplementary figures is necessary. Describe it in Figure legend. (e.g. n=3 independent cultures or n=12 animals). Number of animals and /or cell cultures are now added to the figure legends.

The authors mentioned Fig. 4a (Line 127), Fig. 5a-b (line 130), and Fig. 4b (line 185). The order of Figure description is better if the authors demonstrate it orderly (i.e. Fig. 4a -> Fig. 4b -> Fig. 5a-b). We re-ordered the text according to reviewer´s suggestions to present the figures orderly (page 2 lines 132-6). Moreover, the order of Supplementary Figures 1 and 2 was changed for the same reason.

In Fig. 4b, Fig. 6a-b, and supplementary Figures, what dose 100% represent? 100% represents control untreated cells. We made it clear in the figure legends (lines 154 and 216).

In Sup. Fig. S3b-d, the western blot data does not seem to match with Graph. In Fig. S3d, the first line of band is being cut (not complete band).The graphs are the average values from 6 independent blots while the Western blots are the representative pictures. We chose different pictures to match the average values better and also corrected the cut blot in Fig. S3d.

Figure legend for Fig. S3d: [Supp. Fig. S3. The effect of bilirubin on NF-κB signaling pathway. Primary hepatocytes were pre-incubated with BR (0-40 µM) for 2 h and then treated with TNF-α (12 ng/ml) for 5 min. Total (b) IκBα, (d) IKKβ and phosphorylated (a) IκBα and (b) IKKα/β were measured by Western blot.]. There is no (c) explanation. We corrected this oversight.

Line 343: CO2 -> CO2 We corrected the subscript (line 354).

Reviewer 2 Report

This well written paper investigates the mechanisms behind the anti-inflammatory actions of bilirubin.  Utilizing a Gunn rat model, the author’s first document the blunting of the inflammatory response and then explore possible mechanisms to explain this response.

While the observations and results that BR does blunt the inflammatory response are demonstrated in these experiments, how BR is responsible for this protective effect is further investigated but not totally explained.

Results: 

 Section 2.1.:

The investigators clearly demonstrate the blunting of the LPS induced inflammatory response (which has been demonstrated in several previous publications). 

Section 2.2.:

Both heterozygous and Gunn rat hepatocytes had similar intracellular BR levels yet Gunn rat hepatocytes were more resistant to TNF – α both before and after BR pretreatment.  This raises the question is there something else in these cells other than BR that is protective.

The next portion of this section deals with the induction of LBP mRNA which occurred only in Gunn rat hepatocytes but not heterozygote hepatocytes. This is interesting and discussion of this finding would be of interest. (explanation)

The figures 4b should have the graph labeled “LBP MRNA in primary Gunn hepatocytes.”  Figures S2 should have label “LBP mRNA in primary heterozygote hepatocytes.”

Section 2.3.:

                These experiments show that BR pretreatment diminished the phosphorylation of the NF-kB p65 subunit.  The text mentions both Gunn and heterozygote hepatocytes were treated.  It is not clear from the text or figure 6 whether these are the results for Gunn or heterozygous hepatocytes.  If the subunit was decreased, why weren’t the total levels of NF-kB p65 protein also decreased?

Discussion:  The results section raises several questions.

Paragraph 1:  The authors state data concerning the protective effects of BR from animal models of inflammation is “surprisingly scarce.”  There are certainly many publications using this model and I am not sure scarce is the correct term.

Paragraph 4:  The discussion of balance between pro and anti-inflammatory cytokines is an important one.

Paragraph 5:  Findings concerning LBP are among the most interesting in this study.  Better labeling of the figures 4 and S.2 as previously discussed will be helpful.  The issue of why LBP was only increased in Gunn rat hepatocytes is interesting and should be addressed.

Paragraph 6:  Another interesting finding.  Why there was a decrease in mRNA but not levels of NF-kB protein complex should be addressed.

Paragraph 7:  This is an important question.  If intracellular BR levels in Gunn and heterozygote cells are the same, what are the BR priming mechanisms that result in the blunted inflammatory response?  The authors have demonstrated priming may be the protective mechanism and could expand on this paragraph.

Author Response

Reviewer #2:

Thank you for your positive review and helpful comments. Below see our point-by-point response.

Responses to Reviewer #2:

This well written paper investigates the mechanisms behind the anti-inflammatory actions of bilirubin.  Utilizing a Gunn rat model, the author’s first document the blunting of the inflammatory response and then explore possible mechanisms to explain this response.

While the observations and results that BR does blunt the inflammatory response are demonstrated in these experiments, how BR is responsible for this protective effect is further investigated but not totally explained.

Results: 

 Section 2.1.:

The investigators clearly demonstrate the blunting of the LPS induced inflammatory response (which has been demonstrated in several previous publications). 

Section 2.2.:

Both heterozygous and Gunn rat hepatocytes had similar intracellular BR levels yet Gunn rat hepatocytes were more resistant to TNF – α both before and after BR pretreatment.  This raises the question is there something else in these cells other than BR that is protective. We discuss „bilirubin priming“, adaptive proceses triggered by long lasting hyperbilirubinemia as one of the possible explanations of this phenomenon in the Discussion section of the manuscript (lines 285-289).

The next portion of this section deals with the induction of LBP mRNA which occurred only in Gunn rat hepatocytes but not heterozygote hepatocytes. This is interesting and discussion of this finding would be of interest. (explanation) We discuss these finding in the Discussion section of the manuscript (lines 258-272).

The figures 4b should have the graph labeled “LBP MRNA in primary Gunn hepatocytes.”  Figures S2 should have label “LBP mRNA in primary heterozygote hepatocytes.”We renamed the Figures according to the reviewer´s suggestions.

Section 2.3.:

                These experiments show that BR pretreatment diminished the phosphorylation of the NF-kB p65 subunit.  The text mentions both Gunn and heterozygote hepatocytes were treated.  It is not clear from the text or figure 6 whether these are the results for Gunn or heterozygous hepatocytes.  If the subunit was decreased, why weren’t the total levels of NF-kB p65 protein also decreased? In this experiment, both hepatocytes from heterozygous and Gunn rats were used. As both types of cells displayed similar response to TNFα/bilirubin treatment, the data are presented as a sum of both experiments. We added this information to the Figure 6 legend. Phosphorylation of p65 subunit modulates transcriptional activity of NF-κB independently of inhibitor of κB (IκB) proteins (Hochrainer et al., JBC 2013) thus total p65 protein level and its nuclear translocation do not have to correspond necessarily to the p65 phosphorylation.

 Discussion:  The results section raises several questions.

Paragraph 1:  The authors state data concerning the protective effects of BR from animal models of inflammation is “surprisingly scarce.”  There are certainly many publications using this model and I am not sure scarce is the correct term.  We agree with the reviewer that there are many publications using animal model of inflammation but in fact, there are only few papers using hyperbilirubinemic animals to investigate the role of bilirubin in inflammation and we believe that the sentence „surprisingly scarce data have been published on hyperbilirubinemic animal models of inflammation“ is thus correct.

Paragraph 4:  The discussion of balance between pro and anti-inflammatory cytokines is an important one. Thank you for your comment. 

Paragraph 5:  Findings concerning LBP are among the most interesting in this study.  Better labeling of the figures 4 and S.2 as previously discussed will be helpful.  The issue of why LBP was only increased in Gunn rat hepatocytes is interesting and should be addressed.We re-labeled the figures according to the reviewer´s suggestions. Moreover, we expanded the discussion section on the role of LBP as suggested (lines 261-269).

Paragraph 6:  Another interesting finding.  Why there was a decrease in mRNA but not levels of NF-kB protein complex should be addressed. Thank you for the question but there was probably an oversight. We did not measure NFκB p65 mRNA but the protein levels of both total and phosphorylated NFκB p65 by Western blot. For some cell types, it is characteristic that stimulation with TNF-alpha results in an increase in NF-κB p65 phosphorylation while the concentrations of the total NF-κB p65 protein remain constant (for ilustration, see the Western blot below from Cell Signaling Technology, the antibody provider).

Source: https://www.cellsignal.com/products/primary-antibodies/phospho-nf-kb-p65-ser536-93h1-rabbit-mab/3033

Paragraph 7:  This is an important question.  If intracellular BR levels in Gunn and heterozygote cells are the same, what are the BR priming mechanisms that result in the blunted inflammatory response?  The authors have demonstrated priming may be the protective mechanism and could expand on this paragraph. We agree with the reviewer that this topic is very interesting and important, however, identifying these factors is far beyond the scope of this manuscript and requires an independent study. In fact, we are collaborating on a grant project trying to solve this issue. We added these comments as a future perspective into the Discussion section (lines 287-289).